# Electrospun Microstructured Biopolymer Fibers Containing the Self-Assembled Boc–Phe–Ile Dipeptide: Dielectric and Energy Harvesting Properties

Adelino Handa ⬡, Rosa M. F. Baptista *⬡, Daniela Santos, Bruna Silva ⬡, Ana Rita O. Rodrigues ⬡, João Oliveira, Bernardo Almeida, Etelvina de Matos Gomes and Michael Belsley ⬡

Centre of Physics of Minho and Porto Universities (CF-UM-UP), Laboratory for Materials and Emergent Technologies (LAPMET), University of Minho, Campus de Gualtar, 4710-057 Braga, Portugal; handa.adelino@gmail.com (A.H.); magsantosdaniela@gmail.com (D.S.); mdasilvabruna@gmail.com (B.S.); ritarodrigues@fisica.uminho.pt (A.R.O.R.); jomi0fisica@gmail.com (J.O.); bernardo@fisica.uminho.pt (B.A.); emg@fisica.uminho.pt (E.d.M.G.); belsley@fisica.uminho.pt (M.B.)
* Correspondence: rosa_batista@fisica.uminho.pt

**Abstract:** Hybrid biomaterials were engineered using the electrospinning technique, incorporating the dipeptide Boc–L-phenylalanyl–L-isoleucine into microfibers composed of biocompatible polymers. The examination by scanning electron microscopy affirmed the morphology of the microfibers, exhibiting diameters ranging between 0.9 and 1.8 µm. The dipeptide self-assembles into spheres with a hydrodynamic size between 0.18 and 1.26 µm. The dielectric properties of these microfibers were characterized through impedance spectroscopy where variations in both temperature and frequency were systematically studied. The investigation revealed a noteworthy rise in the dielectric constant and AC electric conductivity with increasing temperature, attributable to augmented charge mobility within the material. The successful integration of the dipeptide was substantiated through the observation of Maxwell–Wagner interfacial polarization, affirming the uniform dispersion within the microfibers. In-depth insights into electric permittivity and activation energies were garnered using the Havriliak–Negami model and the AC conductivity behavior. Very importantly, these engineered fibers exhibited pronounced pyroelectric and piezoelectric responses, with Boc–Phe–Ile@PLLA microfibers standing out with the highest piezoelectric coefficient, calculated to be 56 pC/N. These discoveries help us understand how dipeptide nanostructures embedded into electrospun nano/microfibers can greatly affect their pyroelectric and piezoelectric properties. They also point out that polymer fibers could be used as highly efficient piezoelectric energy harvesters, with promising applications in portable and wearable devices.

**Keywords:** dipeptides; electrospinning; microfibers; dielectric properties; pyroelectricity; piezoelectricity

## 1. Introduction

The research on organic biomaterials has gained significant attention in recent years due to their unique properties [1]. In this work, we investigate the remarkable piezoelectrical properties of the dipeptide (*tert*-butoxycarbonyl)–L-phenylalanyl–L-isoleucine (Boc–Phe–Ile) when incorporated into polymer nanofibers using the electrospinning technique. Organic materials such as dipeptides have increasingly been captivating the scientific community thanks to their distinctive characteristics, the main one being its energy harvesting ability [2–5].

An interesting feature of dipeptides is their ability to self-assemble into various nanostructures. Specifically, linear and cyclic dipeptides derived from the dipeptide L-phenylalanyl–L-phenylalanine (Phe–Phe) exhibit this remarkable property, forming a diverse set of nanostructures, including nanospheres, nanotubes, and nanofibrils [6–13]. This inherent ability to self-assemble is particularly important as it opens the door to the

creation of a wide range of nanostructured materials with potential applications in several fields [14,15].

The electrospinning technique has emerged as a powerful tool for producing fibers from a polymer solution doped with active molecules, namely dipeptides [16–20]. This involves applying a strong electric field between the tip of a syringe and a collector, resulting in the formation of fibers with diameters on the micro or nanometer scale [21,22]. Electrospinning allows precise control of the morphology and molecular orientation of the active molecules and the polymer chains that make up the fibers, which makes it a valuable technique for producing materials with unique properties [16,23,24].

In addition, it is important to highlight the remarkable piezoelectric properties observed in the dipeptides derived from Phe–Phe [4,25–27]. Piezoelectricity enables the conversion of mechanical energy into electrical energy which makes these materials highly valuable for energy harvesting. On the other hand, pyroelectricity involves responding to changes in temperature. Extensive research has explored these properties in dipeptides derived from Phe–Phe, producing impressive results. For example, power generators based on piezoelectric Phe–Phe nanotubes were successfully produced and demonstrated the capability to yield an output voltage reaching 2.8 V along with an associated power output of 8.2 nW when subjected to periodic application of a 42 N force [28]. Additionally, when examining Phe–Phe vertical microrod arrays produced under the influence of an external electric field, researchers were able to ascertain an effective piezoelectric constant, denoted as $d_{33}$, measuring 17.9 pm V$^{-1}$.

Previous studies have meticulously investigated the integration of three Boc-protected analogs of Phe–Phe dipeptides, namely, Boc–PhePhe, Boc-*p*-nitro–L-phenylalanyl-*p*-nitro–L-phenylalanine (Boc-pNPhepNPhe), and Boc–L-phenylalanyl–L-tyrosine (Boc–PheTyr), into micro- and nanostructured PLLA (Poly-L-lactic acid) polymer fibers [29,30]. These studies revealed the self-organization of these dipeptides within the polymer matrix and showcased the generation of substantial piezoelectric voltages through the piezoelectric effect. Notably, the incorporation of dipeptides led to an enhancement of the piezoelectric response of the PLLA polymer which already possesses inherent piezoelectric properties.

The choice of Boc–Phe–Ile as the object of this work is the result of its similarities with the Phe–Phe dipeptide, widely investigated for its piezoelectric properties. Our research focuses on analyzing the resulting variations in dielectric, piezoelectric, and pyroelectric behavior when investigating a dipeptide from the same family but with the substitution of an aromatic side chain amino acid (phenylalanine, Phe) for one with an aliphatic nature (isoleucine, Ile). This study allowed us to explore the influences of the different molecular structures within the dipeptide family on the piezoelectric, pyroelectric, and dielectric properties observed in our hybrid bionanomaterials. Therefore, in addition to improving our understanding of dielectric properties and piezoelectric dynamics in dipeptide microfibers, our research opens up new perspectives for advances in the development of biomaterials and applications in energy harvesting. This progress is crucial for the creation of innovative, environmentally compatible, and sustainable technologies.

## 2. Materials and Methods

### 2.1. Materials

For this work, two distinct polymers were chosen for fiber production: Poly-L-lactic acid (PLLA, with a molecular weight range of 217–225,000) sourced from Corbion in Gorinchem, the Netherlands, and Poly (methyl methacrylate) (PMMA, with a molecular weight of 996,000), acquired from Merck/Sigma-Aldrich (Darmstadt, Germany).

L-Phenylalanyl-L-Isoleucine dipeptide (abbreviated as Phe–Ile) and di-*tert*-butylpyrocarbonate (known as Boc$_2$O or Boc) were acquired from the suppliers Bachem (Muenchenstein, Switzerland) and Alfa Aesar (Erlenbachweg, Kandel, Germany), respectively, and used as received. To dissolve the polymers and dipeptides, various solvents were employed, including dichloromethane (DCM), *N*,*N*-dimethylformamide (DMF), and chloroform, all

sourced from Merck/Sigma-Aldrich (Darmstadt, Germany), and 1,4-dioxane, obtained from Fisher Chemicals (Zurich, Switzerland).

In our study, we conducted the functionalization of the commercial dipeptide Phe–Ile with the Boc group, aiming to enhance its solubility in the selected organic solvents which were also used to dissolve the chosen polymers for this research. The improved solubility of the dipeptide played an important role, enabling the incorporation of a significantly larger quantity into the resulting fibers. This increased incorporation made a significant contribution to the potential enhancement of the electrical properties of the fibers, representing a critical advancement in this study. The functionalization of the Phe–Ile dipeptide was carried out by applying established protecting group chemistry, introducing the conventional Boc (*tert*-butyloxycarbonyl) protecting group [31]. More precisely, we utilized the Boc group to protect the N-terminus of the phenylalanine amino acid of Phe–Ile dipeptide.

Scheme 1 illustrates the process of functionalization. It initiated with the dissolution of 0.5 g of Phe-Ile in a mixture of 1,4-dioxane (5 mL), water (3 mL), and a 1 M sodium hydroxide (NaOH) solution (3 mL). The resulting solution was stirred at 500 rpm and cooled to a temperature of 0 °C using an ice-water bath for 20 min. After achieving complete homogeneity, di-*tert*-butylpyrocarbonate (0.43 g, 1.1 equivalents) was introduced and stirring was maintained at room temperature for 40 h. Following this reaction period, 50 mL of ethyl acetate was added to the mixture, the stirring was continued, and the solution was acidified using a diluted solution of potassium hydrogen sulfate, $KHSO_4$ (6.31 g, 50 mL), to achieve a pH of 2 to 3 (confirmed using Congo red indicator). Two layers were formed and the aqueous layer was rinsed with ethyl acetate. Ethyl acetate was removed from the organic layer through vacuum evaporation, resulting in the formation of a pure white solid of (*tert*-butoxycarbonyl)–L-phenylalanyl–L-isoleucine (Boc–Phe–Ile) dipeptide. This functionalized dipeptide was subsequently subjected to a 24 h drying process in an oven operating at a temperature of 50 °C, yielding 92% and indicating high efficiency of the functionalization.

**Scheme 1.** Scheme for the synthesis of Boc–Phe–Ile dipeptide through functionalization/Boc-protection of the Phe-Ile dipeptide.

To validate the chemical structure of the Boc-protected dipeptide, Boc–Phe–Ile, nuclear magnetic resonance (NMR) spectroscopy was performed using a Bruker Avance III 400 instrument operating at a frequency of 400 MHz for $^1H$. The FTIR-ATR spectrum of Boc–L-phenylalanyl–L-isoleucine powder, acquired using a Spectrum Two™ spectrophotometer from PerkinElmer (Waltham, MA, USA), displayed a characteristic peak at 1711 $cm^{-1}$, confirming the presence of a carbonyl group belonging to the Boc group (see Figure S1). The $^1H$ NMR spectrum exhibited distinct peaks at δ 1.4 ppm (singlet, 9H, $C(CH_3)_3$) and 5.2 ppm (broad singlet, 1H, NH Boc), confirming the successful incorporation of the Boc group (see Figure S2). The incorporation of the Boc groups is crucial as it not only increases the stability of the molecule but also facilitates subsequent functionalization steps, as highlighted in the manuscript. Notably, it promotes the solubility of the dipeptide in the solvents used for preparing polymeric solutions and, in a significant quantity, enhances the physical and electrical properties of the hybrid fibrous material.

### 2.2. Polymeric Solutions and Electrospinning Microfibers

To prepare 5 mL of 10% (*w*/*v*) polymeric solutions for each selected polymer, the following methods were employed: 0.5 g of PLLA was dissolved in 4 mL of dichloromethane (DCM) at 35 °C and the solution was stirred at 600 rpm until complete dissolution. Subsequently, the dipeptide was dissolved in 1 mL of dimethylformamide (DMF). The dipeptide solution was then added to the PLLA solution at a 4:1 (*v*/*v*) solvent ratio while stirring at 300 rpm at room temperature for 24 h. A 10% (*w*/*v*) PMMA polymeric solution was prepared using a similar method and it was conducted at room temperature without the need for temperature adjustments.

Electrospinning was carried out using a 5 mL syringe with a 0.232 mm inner diameter needle (0.5 mm outer diameter) connected to a high-voltage power supply (Spellmann CZE2000, Bochum, Germany). The electrospinning setup was vertically configured and operated at standard room temperature and pressure. The collector, functioning as an electrode, was cylindrical and rotatable.

To produce bead-free fibers, the following critical parameters were considered: the needle-to-collector distance (11 cm), an electric potential difference of 20 kV, and the solution flow rate (0.10–0.20 mL/h). These parameters were carefully controlled to ensure the successful electrospinning of uniform fibers without bead formation (see Figure 1 for a visual representation of the electrospinning setup). Furthermore, after preparation, the samples are subjected to oven drying before measurements.

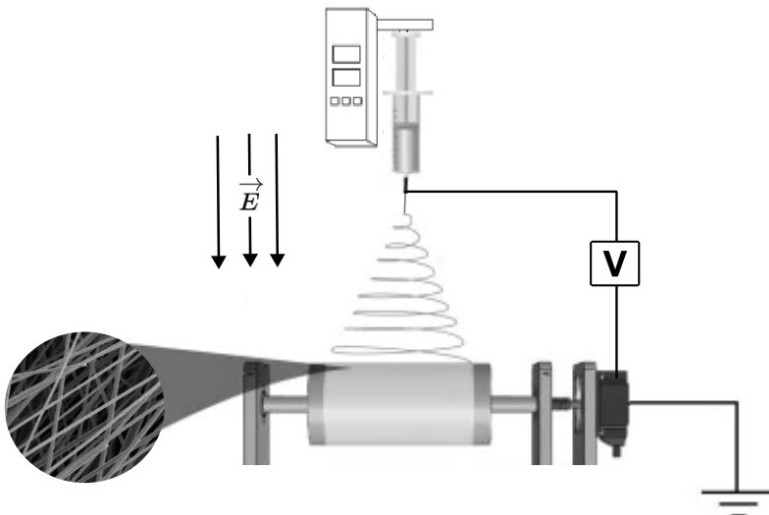

**Figure 1.** Representative electrospinning scheme.

### 2.3. Scanning Electron Microscopy (SEM)

The morphology, size, and shape of both Boc–Phe–Ile structures and the electrospun fibers, with embedded Boc–Phe–Ile dipeptide, were evaluated using Scanning Electron Microscopy (SEM), a Nova Nano SEM 200 with an accelerating voltage of 10 kV. Sample preparation for SEM imaging involved coating the fiber mats with a thin layer (10 nm) of Au-PD (80–20 w%) using a high-resolution sputter coater, specifically the Cressington Company's model 208HR, in conjunction with the Cressington MTM-20 high-resolution thickness controller. The average fiber diameters were determined from SEM images and this analysis was performed using ImageJ software (version 1.41) [32].

### 2.4. Optical Absorption and Photoluminescence

These measurements were crucial for characterizing the optical properties and behavior of the Boc–Phe–Ile samples. The Boc–Phe–Ile solutions and fiber mats were subjected to optical absorption (OA) assessments using a Shimadzu UV-3101PC UV–Vis-NIR spectrophotometer from Shimadzu Corporation in Kyoto, Japan. For photoluminescence evaluations, a Fluorolog 3 spectrofluorimeter from HORIBA Jobin Yvon IBH Ltd. in Glasgow, UK,

was put to use. To perform optical absorption measurements, the Boc–Phe–Ile solutions were prepared in methanol. These samples were introduced into a quartz cuvette with a 1 cm light path. In the case of photoluminescence (PL) measurements, the spectra were captured within the wavelength span of 275–600 nm. An excitation wavelength of 275 nm was deployed with fixed input and output slits to attain a spectral resolution of 2 nm. Photoluminescence excitation (PLE) spectra were also collected, covering the wavelength range of 240–320 nm.

To investigate the reflectance of the fiber mats, a UV-2501PC spectrophotometer equipped with an integrating sphere (ISR-205 240A), sourced from Shimadzu Corporation in Kyoto, Japan, was employed for measuring the diffuse reflectance spectra. Barium sulfate served as the reference material. The measurement covered a wavelength range of 200 to 800 nanometers.

For ascertaining the band gap energy (Eg) of the fiber mats, the Kubelka–Munk function, denoted as $F(R)$, was employed. The Kubelka–Munk function adheres to the relationship expressed as $[h\nu F(R)]^n = \alpha(h\nu - E_g)$ where h$\nu$ denotes the incident photon energy, $\alpha$ represents the absorption coefficient, and $n$ signifies the type of electronic transition. In the context of this study, $n = 1/2$ was utilized given the indirect band gap nature. $F(R)$ is determined based on the material's total reflectance coefficient ($R$) by:

$$F(R_\infty) = \frac{(1 - R_\infty)^2}{2R_\infty} = \frac{K}{S} \tag{1}$$

### 2.5. Dynamic Light Scattering (DLS)

To determine the hydrodynamic size, polydispersity index, and zeta potential of the Boc–Phe–Ile nanostructures, dynamic light scattering (DLS) was employed. Initially, a solution of the dipeptide was dissolved in methanol at a concentration of 0.0006 mg/mL. Subsequently, 100 µL of this solution was added to 2900 µL of ultrapure water. Before the measurements, the solution underwent sonication at 36 °C for 10 min followed by vortex stirring for 2 min. Additionally, it was filtered through a 45 µm pore filter. The measurements were conducted using a quartz cell for assessing the hydrodynamic size and polydispersity index while a glass cell was used for zeta potential measurements, with the measurement taken in backscatter mode.

### 2.6. Dielectric Spectroscopy

Impedance spectroscopy was employed to investigate the dielectric properties of electrospun fibers incorporating self-assembled dipeptide inclusions of Boc–Phe–Ile. This analysis aimed to provide insights into their electrical characteristics, energy storage capabilities, and suitability for various applications. The impedance spectroscopy measurements were conducted within a temperature range of 283–383 K and across a frequency span of 20 Hz to 3 MHz. The complex permittivity, denoted as $\varepsilon$ and expressed as $\varepsilon = \varepsilon' - i\varepsilon''$, where $\varepsilon'$ and $\varepsilon''$ represent the real and imaginary components, respectively, was assessed. The values of $\varepsilon'$ and $\varepsilon''$ were determined based on the measured capacitance (C) and loss tangent (tan δ) using the following equations: $C = \varepsilon'\varepsilon_0(A/d)$ and tan δ $= \varepsilon''/\varepsilon'$. Here, A signifies the electric contact area and d stands for the thickness of the fiber mat.

To conduct these measurements, the samples were configured as parallel plate capacitors and integrated into an LCR network. The aluminum foil, serving as the substrate for collecting the fiber mats, acted as the bottom electrode while the top electrode consisted of a cylindrical metal contact with an approximate diameter of $10^{-2}$ m. Data acquisition was accomplished using a Wayne Kerr 6440A precision component analyzer in conjunction with dedicated computer software. Shielded test leads were employed to minimize parasitic impedances associated with connecting cables. Temperature-dependent measurements were performed at a controlled rate of 2 °C/min, facilitated by a Polymer Labs PL706 PID controller and a furnace.

### 2.7. Pyroelectric Coefficient

Pyroelectricity is a phenomenon arising from the temperature-dependent nature of spontaneous polarization (Ps) in materials lacking centrosymmetry. The pyroelectric coefficient, denoted as p and defined as the rate of change in spontaneous polarization with temperature (dp/dT), characterizes this behavior. To detect alterations in polarization, we measured the pyroelectric current, represented as I, which is directly proportional to the rate of change in polarization. For this purpose, a Keithley 617 electrometer from Keithley Instruments GmbH in Landsberg, Germany, was utilized.

The formula employed to compute the pyroelectric current is given by I = A(dp/dT)(dT/dt) where A denotes the electrode's surface area and dT/dt represents the rate of temperature change. These measurements were carried out within a capacitor configuration, specifically under short-circuit conditions. The fiber mat sample, with an area of $\pi \times 52$ mm$^2$ and thickness ranging from 20 to 330 μm, was arranged as a parallel-plane capacitor.

### 2.8. Piezoelectric Measurements

The microfiber mats were directly adhered to high-purity aluminum sheets, functioning as the electrodes for piezoelectric assessments. These resultant specimens were firmly situated on a platform and uniform perpendicular pressures were applied across the entire surface of each specimen. The assessed fiber mat arrays possessed dimensions of $(30 \times 40)$ mm$^2$, with thickness fluctuating between 20 and 160 μm. They were subjected to cyclic mechanical forces, induced by a vibration emitter Frederiksen SF2185 (Frederiksen Scientific A/S, Olgod, Denmark), operating at an amplitude of 500 mVpp and a frequency of 3 Hz, supplied by a signal generator (Hewlett Packard 33120A, HP Inc., Palo Alto, CA, USA).

Prior to the experiment, the exerted forces were standardized via a force-sensing resistor (FSR402, Interlink Electronics Sensor Technology, Graefelfing, Germany). The resultant piezoelectric voltage output was gauged employing a 100 MΩ resistor linked to a 6 dB low-frequency filter and succeeded by a low-noise preamplification unit (research system SR560). Subsequently, the voltage readings were logged with a digital storage oscilloscope (Agilent Technologies DS0-X-3012A, Agilent Technologies, Santa Clara, CA, USA). To determine the charge, Q, the applied force time interval, $\Delta t$, and the maximum piezoelectric current, $I_{max}$, the following equation was employed: $Q = I_{max}\Delta t$. The effective piezoelectric coefficient, denoted as $d_{eff}$, was calculated by considering the charge Q and the applied force $F_{app}$, utilizing the equation $d_{eff} = Q/F_{app}$.

## 3. Results and Discussion

### 3.1. Morphology of Self-Assembled Dipeptides in Solution and in Fibers

The dipeptide Boc–Phe–Ile undergoes self-assembly into microspheres (NS) with an average diameter of $1.7 \pm 0.4$ μm when dissolved in a methanol solution. These microspheres were illustrated in Figure 2($a_1$,$a_3$). Very interestingly, the microspheres, as observed through SEM analysis, are formed by smoothly layered surfaces.

For the Boc–Phe–Ile@PLLA fibers, from the 62 measurements performed, an average diameter of $0.87 \pm 0.24$ μm with a maximum value of 1.51 μm was calculated, as shown in Figure 2($b_2$). The fibers are well oriented and defined, presenting some porosity on their surface (Figure 2($b_1$,$b_3$), respectively).

For the Boc–Phe–Ile@PMMA sample, 42 measurements were performed. The histogram of the images obtained by SEM, shown in Figure 2$c_2$, has an average diameter of $1.85 \pm 0.60$ μm, with a maximum value of 3.07 μm. Along the entire length of these fibers, it is possible to see the presence of porosity. The formation of pores in fibers depends on several factors, such as the manufacturing process, the properties of the polymer, and the processing conditions.

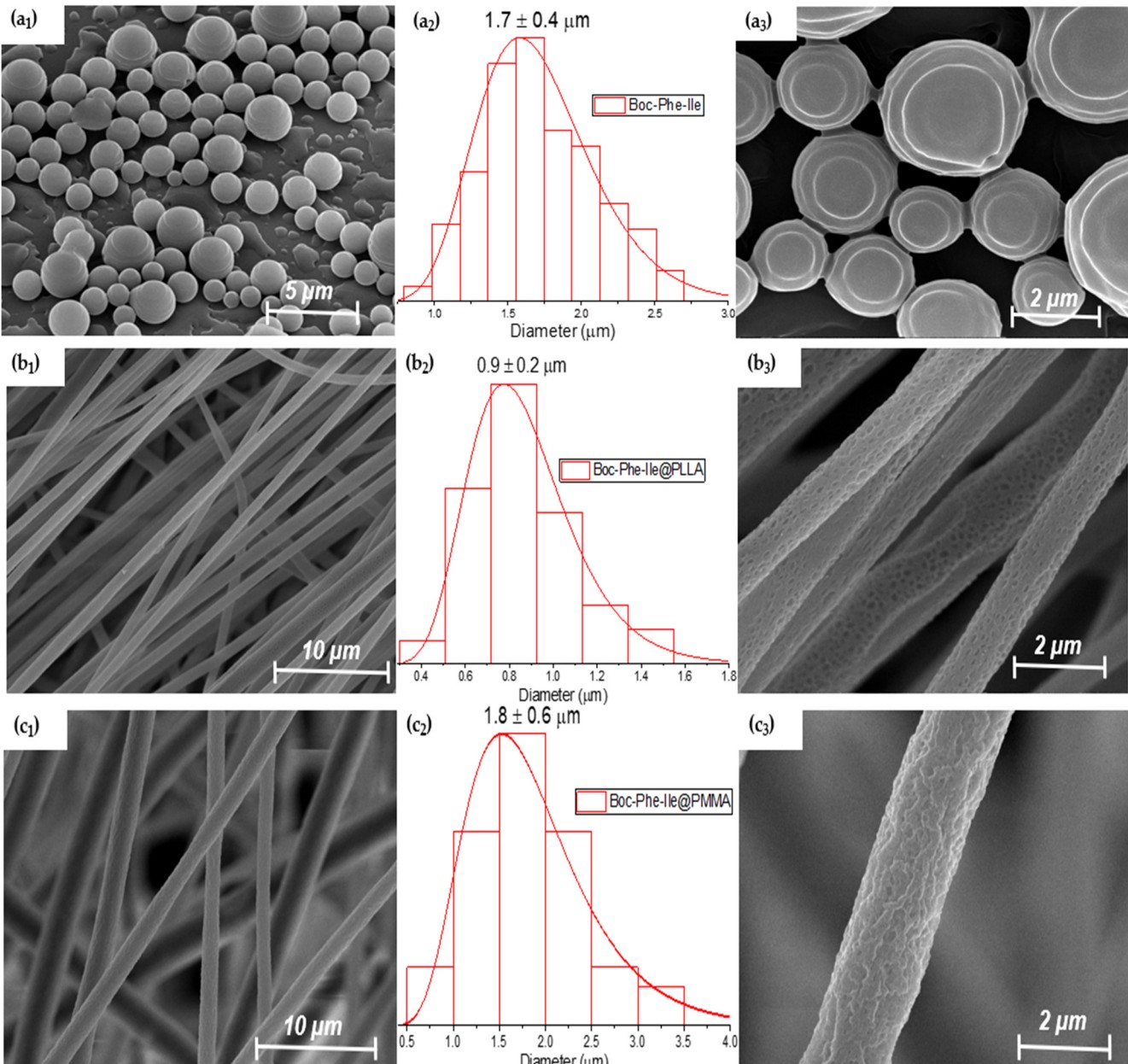

**Figure 2.** (**a₁**–**a₃**) SEM images of dipeptide microspheres for Boc–Phe–Ile solutions with magnifications of 15,000× and 50,000×; SEM images with magnifications of 5000×, 10,000×, and 50,000× and their respective distributions for histograms PLLA (**b₁**–**b₃**) and PMMA (**c₁**–**c₃**) with the embedded Boc–Phe–Ile dipeptide.

In order to improve our comprehension of the dipeptide's self-assembling process and draw comparisons with results obtained through scanning electron microscopy (SEM), we conducted measurements using the dynamic light scattering (DLS) technique.

It is important to emphasize that DLS provides insights into the hydrodynamic size, specifically the size of particles in motion. The DLS study revealed the presence of two populations of microspheres: one with a smaller percentage of microspheres, having an average hydrodynamic diameter of 0.18 μm, and a second population that comprises the majority of microspheres, exhibiting an average hydrodynamic diameter of 1.26 μm, as illustrated in Figure 3. The presented results emphasize a remarkable consistency in the size measurements of these microspheres, as determined by both scanning electron microscopy

(SEM) and dynamic light scattering (DLS). The SEM-derived average diameter of 1.7 μm aligns perfectly with the DLS-measured average hydrodynamic size of 1.26 μm.

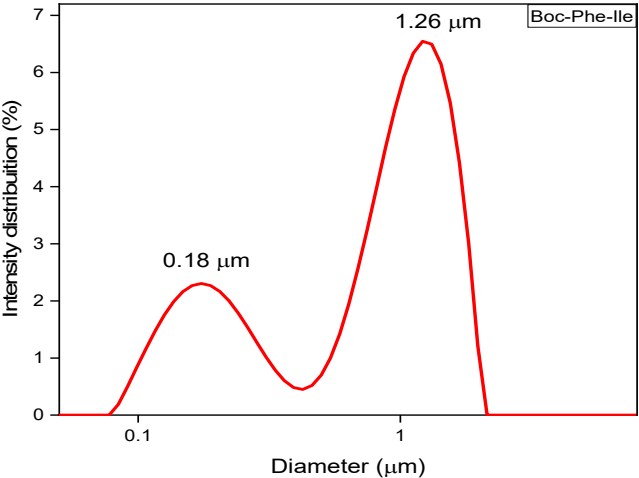

**Figure 3.** Intensity weighted particle size distributions for Boc–Phe–Ile microspherical structures measured by dynamic light scattering.

Moreover, the determined value of the zeta potential for Boc–Phe–Ile microspheres, which stands at $(-24.62 \pm 0.52)$ mV, indicates that the molecules remain stable in an aqueous medium and serve as evidence of their polarity. Furthermore, the high transmittance values (86.01%) and the low polydispersity index (0.29%) suggest promising applications for these microspheres in biomedicine, notably for drug delivery [33] as well as in biomarker production.

### 3.2. Optical Absorption and Photoluminescence of Dipeptide Self-Assemblies

The absorption measurements represented in Figure 4a reveal bands within the spectral region of 240–280 nm. These bands manifest at wavelengths of 247 nm (5.02 eV), 253 nm (4.91 eV), 258 nm (4.80 eV), 264 nm (4.70 eV), and 267 nm (4.64 eV), with an average spacing of approximately 4 nm between adjacent peaks.

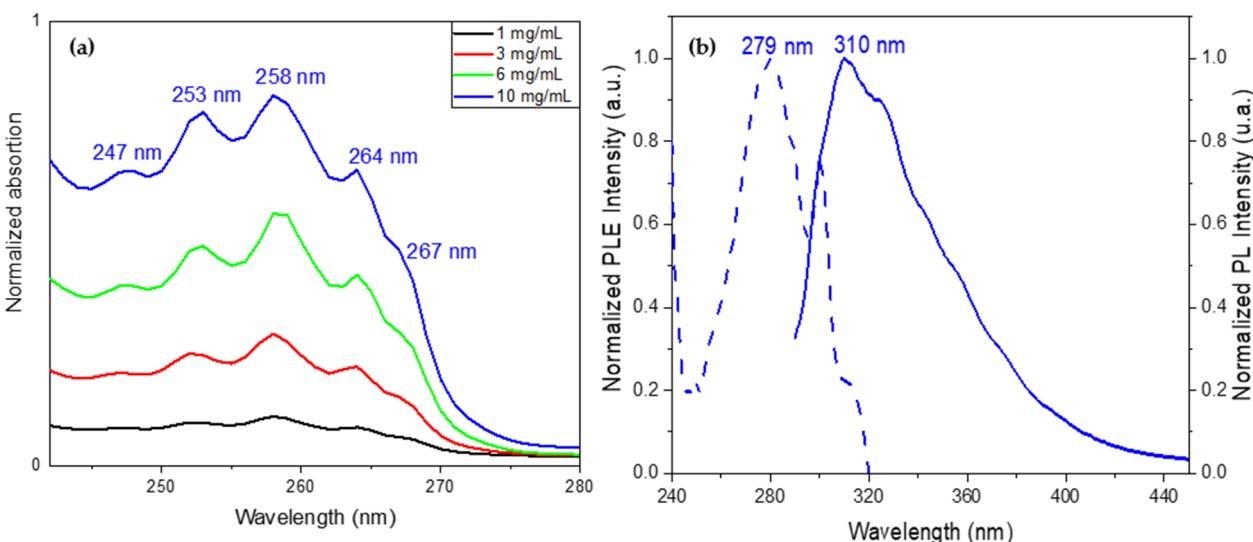

**Figure 4.** In MeOH, (**a**) normalized UV–vis absorption spectra at different solution concentrations of the Boc–Phe–Ile dipeptide and (**b**) normalized excitation and emission spectra.

The presence of these absorption peaks, when normalized with respect to the spectrum of the highest concentration, strongly implies the formation of self-assembled microspheres

exhibiting a quantum confinement effect [30]. As expected, an increase in concentration corresponds to a heightened intensity of absorption.

The excitation spectrum displays a band encompassing wavelengths similar to those observed in the absorption spectrum, with the peak intensity notably concentrated in the range between approximately 270 nm and 290 nm. This observation underscores that the light emitted in the emission spectrum, with its peak intensity ranging from 310 nm to 360 nm, originates from electronic transitions corresponding to the absorption bands, Figure 4b.

The graph of the Kubelka–Munk function is presented and determined the energy of the band gap in the Figure 5 inset for the Boc–Phe–Ile@PLLA with Eg = 4.51 ± 0.01 eV and the Boc–Phe–Ile@PMMA with Eg = 4.43 ± 0.01 eV.

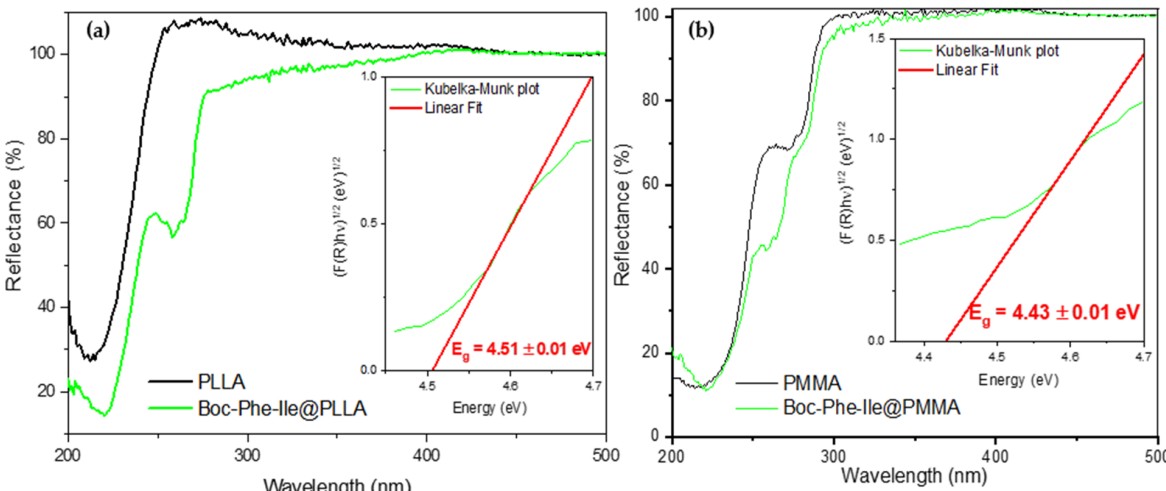

**Figure 5.** Reflectance spectra of (**a**) Boc–Phe–Ile@PLLA and (**b**) Boc–Phe–Ile@PMMA fibers. The inset displays the band gap energy determined through the Kubelka–Munk function indicated by the linear fit represented in red.

The analysis of the emission spectra of the Boc–Phe–Ile microspheres in solution, Figure 4b, and Boc–Phe–Ile@PMMA microfibers dissolved in DCM, Figure 6, confirms the existence of the Boc–Phe–Ile dipeptide within the fiber mats, as evidenced by their emission maximum occurring at close wavelengths. Upon excitation of the fibers, a blue photoluminescence, coupled with a noticeable 21 nm redshift of the spectrum (referred to as batochromic displacement), is observed.

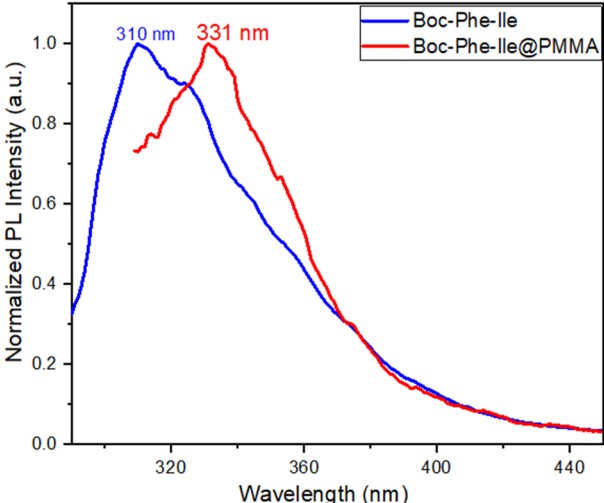

**Figure 6.** Photoluminescence spectra from Boc–Phe–Ile in methanol solution (in blue) and when embedded into PMMA fibers, after polymer fiber dissolution in DCM (in red).

### 3.3. Dieletric Properties

The variation in electrical permittivity with frequency was determined for the composite polymeric nanofibers containing Boc–Phe–Ile dipeptide inclusions, spanning from room temperature to 368 K. Figure 7a–d illustrates the actual and imaginary aspects of the dielectric permittivity for specimens featuring Boc–Phe–Ile@PLLA and Boc–Phe–Ile@PMMA fibers, respectively.

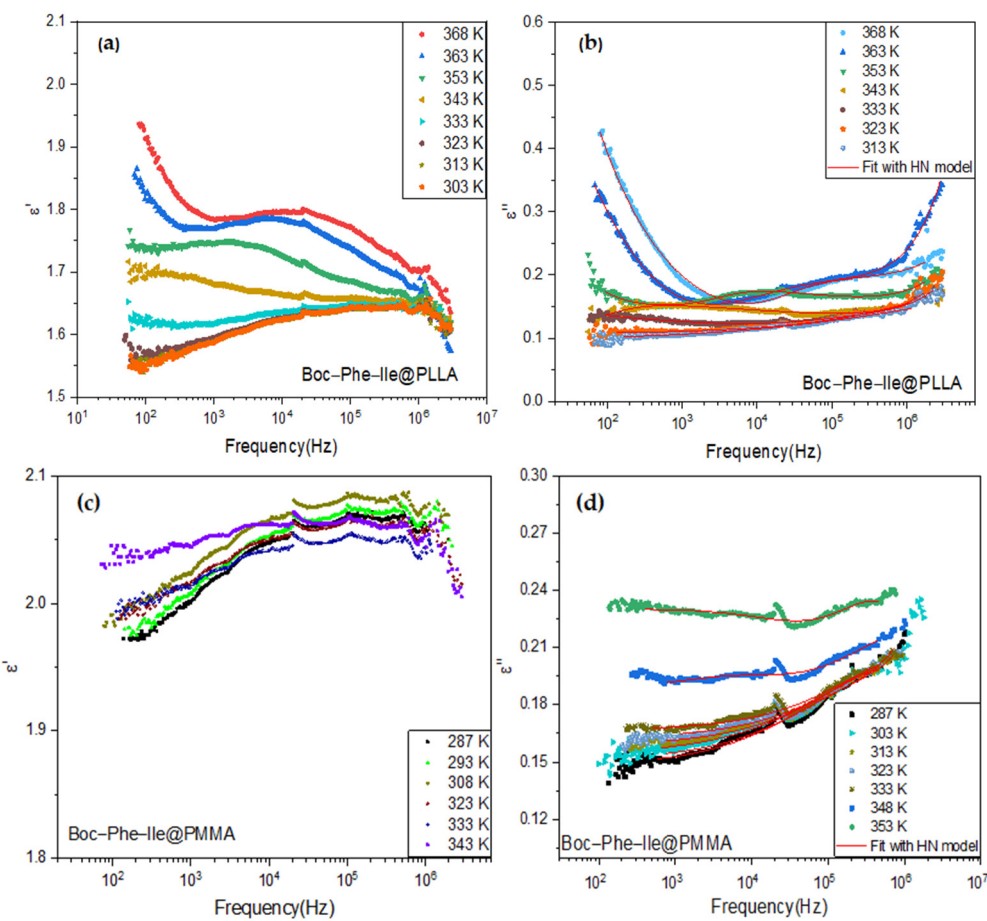

**Figure 7.** Frequency dependence of the real and the imaginary part of the dielectric permittivity, measured at different temperatures on the (**a**,**b**) Boc–Phe–Ile@PLLA and (**c**,**d**) Boc–Phe–Ile@PMMA.

The real and imaginary components of the permittivity of both samples, with the different polymeric matrices, have small variations in the whole temperature interval. Nevertheless, a difference is observed in their low frequency region. For the sample Boc–Phe–Ile@PLLA, there is an evident rise in $\varepsilon'$ as the frequency decreases. This phenomenon can be attributed to Maxwell–Wagner interfacial polarization [34] which arises from the presence of interfaces within the material, a consequence of the dipeptides being integrated into the fibers. In fact, Maxwell–Wagner polarization arises on materials presenting a heterogeneous character, such as in biological materials, phase separated polymers, composites, blends, crystalline or liquid crystalline polymers, multilayer films, biphasic nanostructures, and others [34]. Due to their composite nature, they give rise to the presence of inner dielectric boundary layers at interfaces or grain-boundaries, from where charges can move or be blocked. As such, when an electric field is applied, charges at interfaces move in response to it, giving rise to interfacial dipoles due to the induced charge separation. These charges can become very far from each other so that the interfacial polarization can reach high values as compared to other polarization mechanisms [34]. This then causes an increase in $\varepsilon'$ with decreasing frequency, as observed in our samples, thus confirming the successful inclusion of the dipeptides inside the nanofibers polymeric matrix.

In the case of purely electronic conductivity, the electric permittivity is primarily imaginary and can be expressed as $\varepsilon'' = \sigma_{DC}/(\varepsilon_0\omega)$ [35] where $\varepsilon_0$ denotes the vacuum dielectric permittivity, $\omega$ represents the angular frequency, and $\sigma_{DC}$ stands for the DC conductivity. However, when dealing with ionic charge carriers, which lead to electrode or Maxwell–Wagner polarization effects, this equation can be generalized. In this scenario, the electric conductivity contribution can be described by the equation $\varepsilon'' = \sigma_{DC}/(\varepsilon_0\omega^s)$ where s is an exponent and $s \leq 1$ [35]. As the conductivity term results in elevated values of the imaginary component of permittivity at low frequencies (as $\omega \to 0$), the low frequency increase in $\varepsilon''$ is due to the electric conductivity contribution to the permittivity. Also, the enhancement of this low frequency increase in permittivity with increasing temperature reflects the corresponding increase in the sample conductivity.

This low frequency increase in the permittivity is not observed for the composite fibers with PMMA polymeric matrix. This is due to the higher resistivity of PMMA ($\sim 10^{16}$ $\Omega$ cm [36]) as compared to PLLA ($\sim 10^8$ $\Omega$ cm [37]). As such, to observe the conductivity-induced rise in the electric permittivity, we would have to go to lower frequencies beyond the measured region.

For Boc–Phe–Ile@PLLA, in Figure 7, we can discern the presence of a peak associated with dielectric relaxation and the onset of second relaxation occurring at higher frequencies more visible in the imaginary component of the electric permittivity. For Boc–Phe–Ile@PMMA, the relaxation is marked by a peak in the real part of the electric permittivity and an increase in $\varepsilon''$ with increasing frequency, which indicates a maximum just outside the measurement window. The observed relaxations are broad, signifying the existence of a range of relaxation times and the manifestation of correlated dipole phenomena. Thus, to fit the electrical permittivity curves and to describe the relaxation behavior, the Havriliak–Negami (HN) model function [35,38] was considered:

$$\varepsilon(\omega) = \varepsilon_\infty + \frac{\Delta\varepsilon}{\left[1 + (i\omega\tau)^\beta\right]^\gamma} \qquad (2)$$

where $\Delta\varepsilon$ is the intensity of the relaxation, $\varepsilon_\infty$ is the high-frequency dielectric constant, $\omega$ is the angular frequency, $\tau$ is the relaxation time, and $\beta$ and $\gamma$ are coefficients of the HN function with the constraints $0 < \beta \leq 1$ and $0 < \beta\gamma \leq 1$. For $\beta = \gamma = 1$, the HN model reduces to the Debye function with only one relaxation time. To take into account the effect of the conductivity, the imaginary part of the permittivity was fitted with Equation (1) plus the conductive term $\varepsilon'' = \sigma_{DC}/(\varepsilon_0\omega)$. The fits are shown in Figure 7b,c for Boc–Phe–Ile@PLLA and Boc–Phe–Ile@PMMA and, from them, the corresponding function parameters were determined.

The temperature dependence of the relaxation time obtained from the fits show characteristic Arrhenius-like behavior, with the dependence $\tau = \tau_0 e^{E_a/k_B T}$ where $k_B$ is the Boltzmann constant, T is the temperature, $\tau_0$ is a constant, and $E_a$ is the activation energy. In order to determine $E_a$, the logarithm of the relaxation time is represented in Figure 8 as a function of the inverse of the temperature so that the slope of the fit gives the activation energy of the relaxation. The obtained activation energies are 1.91 eV for Boc–Phe–Ile@PLLA and 0.53 eV for Boc–Phe–Ile@PMMA. The activation energy of the Boc–Phe–Ile@PLLA fibers is above 1 eV and is characteristic of ionic movement in the sample. This reflects the fact that the PLLA glass transition temperature occurs inside the measured temperature interval ($T_g$ = 323–338 K) [39], leading to ionic displacements inside the polymer as its structure reconstructs around $T_g$.

On the other hand, the glass transition temperature for PMMA is $\sim$378 K [40] which is substantially higher than in PLLA and is beyond the measured temperature region. In this respect, in bulk PMMA for $T < T_g$, the behavior of its electric permittivity is influenced by the $\beta$-relaxation. This relaxation in PMMA arises from the rotation of the pendant side chain (-COOCH$_3$) around the carbon–carbon bond linking the side chain to the polymer backbone [41]. In PMMA, it is known that the beta relaxation dependence with frequency becomes somewhat broad and less intense and its peak shifts to higher frequencies when

PMMA is reduced in size [42]. This is due to chain confinement that induces changes in the equilibrium conformation statistics and orientation distribution of the polymer chains [43]. Since the chain confinement results in more elongated chains in the nanofibers, the local molecular packing is improved which tends to speed up the beta-process [42,43]. From the literature, the β-relaxation activation energy for bulk PMMA is $E_a \sim 0.81$ eV [44] but its value can reach as low as 0.34 eV in thin PMMA films [42]. These values are similar to the $E_a = 0.53$ eV that we obtained in our Boc–Phe–Ile@PMMA fiber mats indicating the presence of the β-relaxation in the fibers due to the PMMA polymeric matrix. Additionally, as the movement of the ionic species in the polymer is hindered, the obtained activation energy can also reflect the behavior of the dipeptide inclusions. Namely, it can also be due of the ionic transitions of the carboxylic acid group at the termination of the dipeptide molecule.

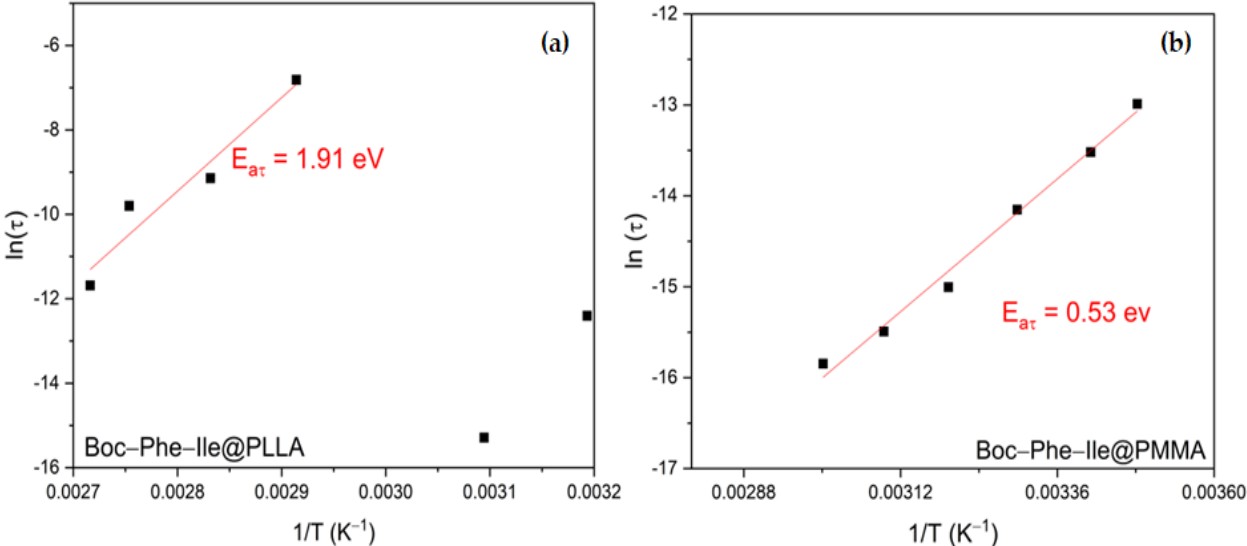

**Figure 8.** Activation energies for the relaxation time contribution obtained from the fits to the imaginary component of the electrical permittivity for (**a**) Boc–Phe–Ile@PLLA and (**b**) Boc–Phe–Ile@PMMA fibers.

To further characterize the electric properties of the composite fibers, their AC electrical conductivity ($\sigma_{AC}$) was studied in the same temperature interval. The AC conductivity is obtained from the imaginary component of the electrical permittivity $\varepsilon''$ through the transformation $\sigma_{AC} = \omega \varepsilon_0 \varepsilon''$ where $\omega$ is the angular frequency and $\varepsilon_0$ is the vacuum permittivity. Figure 9 shows the frequency dependent AC conductivity for (a) Boc–Phe–Ile@PLLA and (b) Boc–Phe–Ile@PMMA. At low frequencies, the AC conductivity is approximately constant, reflecting the presence of the DC conductivity. On the other hand, above a threshold frequency value, $\omega_p$, there is a strong increase in AC conductivity for all samples. To study the factors that govern this behavior, $\sigma_{AC}$ was studied in the framework of the Jonscher power-law dependence [45]:

$$\sigma_{AC} = \sigma_{DC} \left[ 1 + \left( \frac{\omega}{\omega_p} \right)^n \right] \tag{3}$$

where $\sigma_{DC}$ is the DC conductivity, n is an exponent which represents the extent of interaction between mobile ions and their surrounding environment, and $\omega_p$ is the onset frequency for AC-dependent conductivity [46].

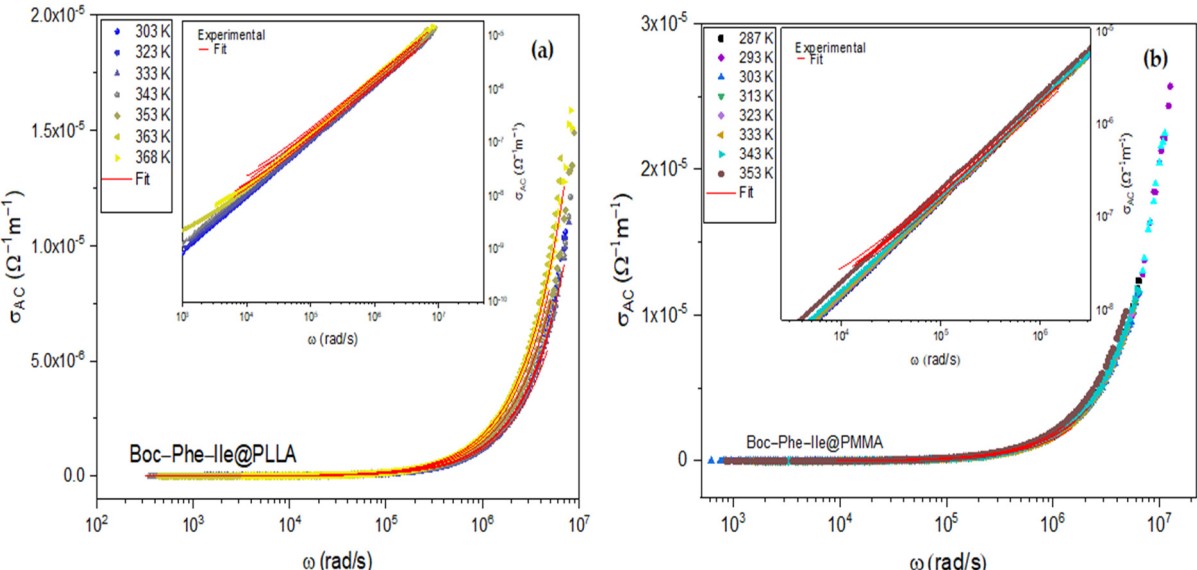

**Figure 9.** AC conductivity as a function of angular frequency for different temperatures of the (**a**) Boc–Phe–Ile@PPLLA and (**b**) Boc–Phe–Ile@PMMA fiber mats. The insets show the corresponding curves on a logarithmic scale, along with the fits with Equation (2).

The insets of Figure 9 show the fitted frequency-dependent $\sigma_{AC}$ curves, with Equation (2), along with the experimental curves. From these fits, the parameters n, $\omega_p$, and $\sigma_{DC}$ were determined. The obtained DC conductivity closely aligns with the values obtained from the imaginary component of the electric permittivity (Figure 7) while the determined n values are consistent with expectations, remaining below 1. On the other hand, Figure 10 shows the logarithm of $\omega_p$ as a function of the inverse of the temperature for samples Boc–Phe–Ile@PLLA and Boc–Phe–Ile@PMMA. As shown, the temperature dependence of the onset frequency follows an Arrhenius behavior, $\omega_p = \omega_{p0}e^{-E_{ap}/k_BT}$, with activation energy $E_{ap}$.

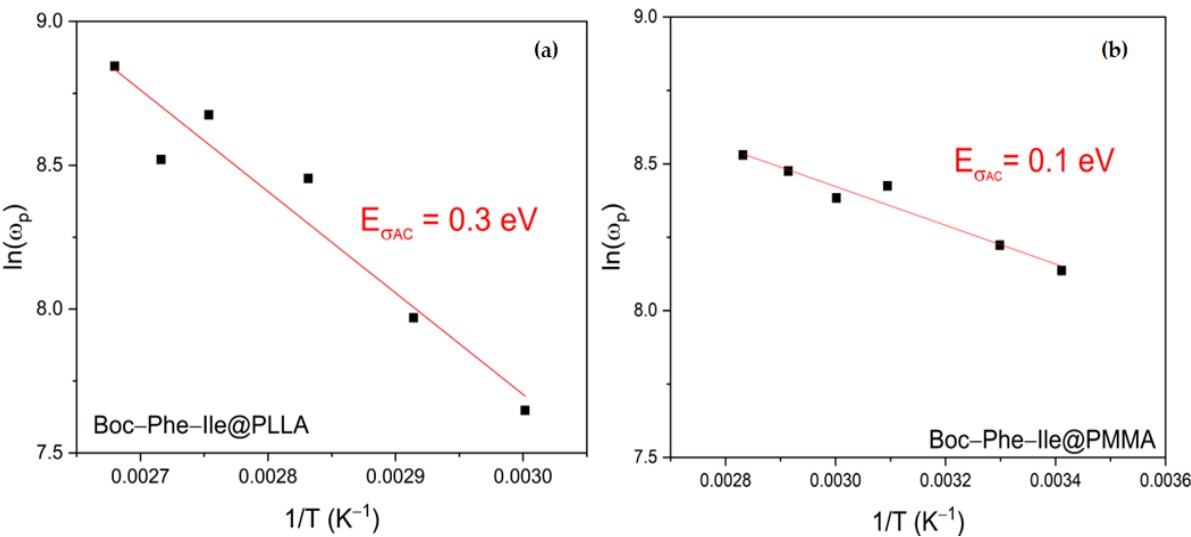

**Figure 10.** Logarithm of the onset frequency $\omega_P$ as a function of the inverse of the temperature for (**a**) Boc–Phe–Ile@PPLLA and (**b**) Boc–Phe–Ile@PMMA fiber mats. Also shown are fits that were used to determine the activation energies.

The obtained activation energies are $E_a$ = 0.30 eV for Boc–Phe–Ile@PPLLA and $E_a$ = 0.1 eV for Boc–Phe–Ile@PMMA fiber mats. These values are characteristic of the

presence of polaron conduction in the composite fibers due to doping or defects in the polymeric matrices [47,48], originated by the presence of the embedded dipeptide inclusions.

### 3.4. Pyroelectricity in Fibers

Based on our comprehensive analysis of the experimental data used to determine the pyroelectric coefficient, we have identified a consistent pattern in the material's response. At lower temperatures, the material demonstrates stable polarization with minimal variation. However, as temperatures increase above the room temperature, there is an increase in the pyroelectric current.

In Figure 11a, we present the pyroelectric response of the Boc–Phe–Ile@PLLA microfibers where a pyroelectric coefficient of $0.3 \times 10^{-6}$ C/(m$^2$K) is attained when the temperature reaches 321 K. In Figure 11b, we present the pyroelectric coefficient of the Boc–Phe–Ile@PLLA sample, achieving its maximum value of $0.1 \times 10^{-6}$ C/(m$^2$K) at 337 K. For the dipeptide embedded in both PMMA and PLLA polymer fibers, the calculated pyroelectric coefficient has a similar magnitude in the order of $10^{-7}$ C/(m$^2$K) at slightly above room temperature. Therefore, we conclude that these fibers could be integrated in thermal sensing devices based on the pyroelectric effect.

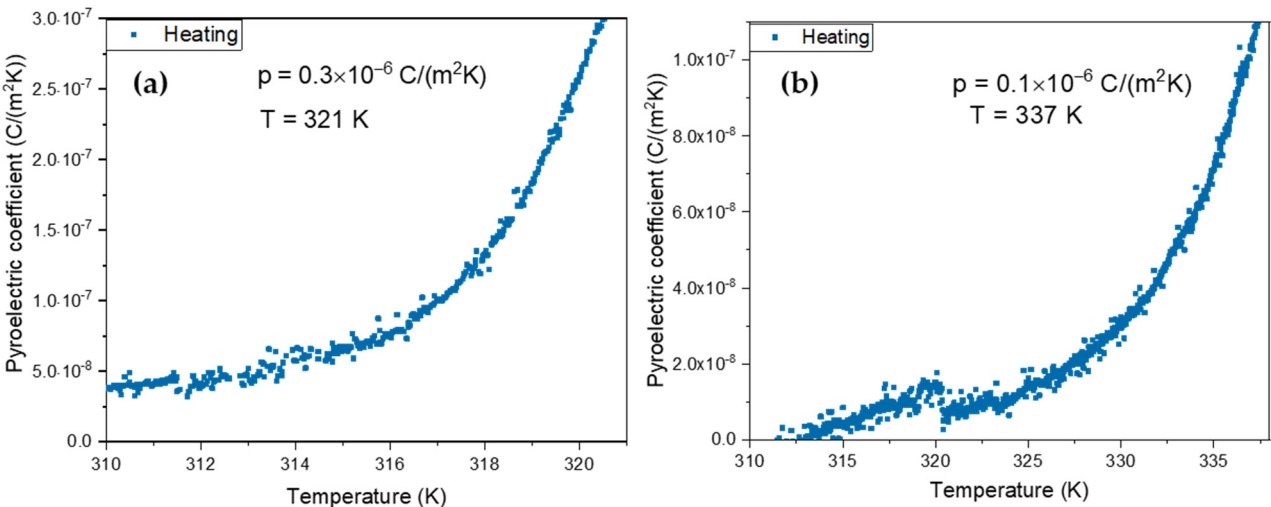

**Figure 11.** Pyroelectric coefficient as a function of temperature of (**a**) Boc–Phe–Ile@PLLA and (**b**) Boc–Phe–Ile@PMMA fiber mats.

### 3.5. Effective Piezoelectric Coefficient

In Figure 12a,b, we can observe the graphical representation of piezoelectric current and output voltage as a function of time for the Boc–Phe–Ile@PLLA and Boc–Phe–Ile@PMMA fiber mats. It is noteworthy that an increase in the applied force results in a proportional increase in both current and output voltage. Furthermore, it is important to emphasize that the intensity of the output signal for each level of applied force remains consistently stable over time, showing minimal variations. A linear regression analysis was conducted to validate the correlation between the current and the applied force, as depicted in Figure 12c. The linear regression results clearly demonstrate a positive and linear pattern, underscoring the direct connection between the applied force and piezoelectric current. This confirms that an increase in force results in a proportional increase in piezoelectric current.

The effective piezoelectric coefficient for the Boc–Phe–Ile@PLLA microfibers was determined for each level of applied force, resulting in an average value of 56 pCN$^{-1}$. In the case of Boc–Phe–Ile@PMMA microfibers, the measured average effective piezoelectric coefficient was 20 pCN$^{-1}$.

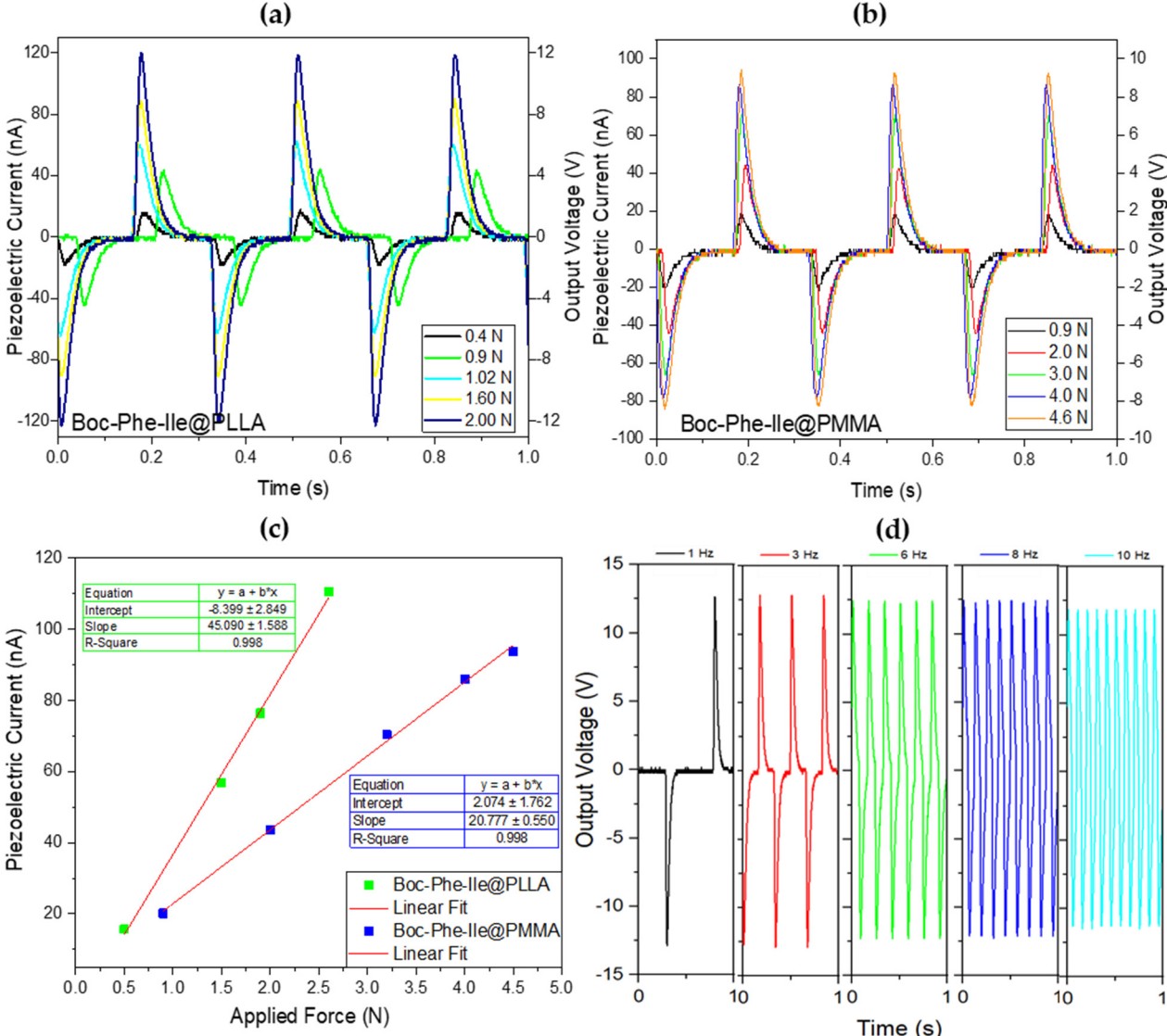

**Figure 12.** (**a**,**b**) Piezoelectric current and output voltage plotted over time; (**c**) maximum piezoelectric current in relation to various periodic forces applied to Boc–Phe–Ile@PLLA and Boc–Phe–Ile@PMMA microfibers; (**d**) variation in the output voltage for Boc–Phe–Ile@PLLA microfibers at different frequencies.

It is worth highlighting that in the Boc-Phe-Leu@PMMA sample, the generation of piezoelectric current and output voltage is primarily driven by the dipeptide molecule Boc–Phe–Ile. The PMMA polymer itself lacks inherent piezoelectric properties.

The higher effective coefficient observed in Boc–Phe–Ile@PLLA can be attributed to the intrinsic piezoelectric nature of the PLLA polymer which possesses an effective coefficient of around ~10 pCN$^{-1}$, as evidenced in prior research [49]. Consequently, the incorporation of the dipeptide clearly enhances the piezoelectric response of the fibers containing the PLLA polymer.

To assess the performance of a Boc–Phe–Ile@PLLA fiber mat, we conducted an analysis of the output voltage under varying frequencies—specifically, 1, 3, 6, 8, and 10 Hz—while maintaining a consistent applied force throughout the same time period, as illustrated in Figure 12d. Notably, the output voltage exhibited stability across all tested frequencies for these fibers.

An essential parameter to consider for the potential application of these nanofiber mats as energy harvesters is the piezoelectric voltage coefficient, denoted as g$_{eff}$ and calculated as d$_{eff}$/($\varepsilon'\varepsilon_0$) VmN$^{-1}$. This figure of merit quantifies the material's suitability as

a piezoelectric sensor. Additionally, it is crucial to highlight another important quantity: the power density released by the nanofiber mat, represented as W and determined by $W = RI^2/A$ ($\mu Wcm^{-2}$), where R equals 100 M$\Omega$ (ohms) for resistance and A denotes the electrode area. The computed values for these parameters are detailed in Table 1.

**Table 1.** Characteristics of the piezoelectric nanogenerator across multiple nanofiber mats.

| Nanogenerator | $d_{eff}$ $pCN^{-1}$ | Force/Area $(Nm^{-2})$ | $g_{eff}$ $(VmN^{-1})$ | Power Density $(\mu Wcm^{-2})$ | Ref. |
|---|---|---|---|---|---|
| Boc–Phe–Ile@PLLA (fiber mat) | 56 | $2 \times 10^3$ | 3.9 | 0.12 | This work |
| Boc–Phe–Ile@PMMA (fiber mat) | 20 | $4 \times 10^3$ | 1.1 | 0.07 | This work |
| Cyclo(L-Trp–L-Trp)@PLLA | 57 | $3 \times 10^3$ | 4.7 | 0.18 | [19] |
| Boc-PhePhe@PLLA (fiber mat) | 8.4 | $4 \times 10^3$ | 0.3 | 2.3 | [29,30] |
| Boc-PheTyr@PLLA (fiber mat) | 7 | $4 \times 10^3$ | 0.3 | 1.0 | [29] |
| Boc-*p*NPhe*p*NPhe@PLLA (fiber mat) | 16 | $4 \times 10^3$ | 0.6 | 9.0 | [29] |

## 4. Conclusions

Composite microfibers were fabricated using the electrospinning technique where the linear chiral dipeptide Boc–Phe–Ile was incorporated into a polymeric matrix comprising different base polymers, namely PLLA and PMMA. Scanning electron microscopy analysis revealed the uniform, aligned, and well-oriented nature of the microfibers, with diameters falling within the range of 0.9 to 1.8 µm. It is worth noting that the choice of solvents for preparing the polymeric solution presented a limitation in the process. The selected solvent had to be compatible with both the polymer and dipeptide, ensuring that both could be dissolved uniformly. This solvent requirement was critical to allow the formation of self-assembled structures during the electrospinning process.

The determination of bandgap energies for both Boc–Phe–Ile and the composite fibers yielded values within the range of 4.4–4.5 eV, indicative of their behavior as (bio)organic semiconducting dielectrics. Investigation into the dielectric properties encompassed temperature and frequency variations, revealing a noticeable increase in the dielectric constant when increasing the temperature. This phenomenon was attributed to the increased availability of energy, facilitating the movement of charges in response to the applied alternating electric field.

Moreover, the presence of Maxwell–Wagner interfacial polarization, particularly on the PLLA-based nanofibers, confirmed the successful integration of the peptide within the fibers. The application of the Havriliak–Negami model to the imaginary component of electric permittivity as a function of frequency allowed for the determination of the ionic hopping activation energies. Furthermore, from the modeling of the AC conductivity, the presence of polaron conductivity was put in evidence, associated with doping or defect formation, due to the successful integration of the dipeptide inside the polymeric fibers.

Finally, and very importantly, this study also revealed interesting pyroelectric and piezoelectric responses of these composite materials. Notably, Boc–Phe–Leu@PLLA nanofibers exhibited a high effective piezoelectric coefficient, reaching 56 pC/N in comparison to other Boc-protected linear dipeptides. These collective findings underscore the capacity of polymeric microfibers to serve as efficient nanogenerators of piezoelectric energy. This capability opens up opportunities for their application in a wide range of portable and wearable devices and health monitoring systems.

**Supplementary Materials:** The following supporting information can be downloaded at: https://www.mdpi.com/article/10.3390/su152216040/s1.

**Author Contributions:** Conceptualization, A.H., R.M.F.B. and E.d.M.G.; investigation, A.H., R.M.F.B., D.S., B.S., A.R.O.R., B.A. and J.O.; writing—original draft preparation, A.H., R.M.F.B. and B.A.; writing—review and editing, R.M.F.B., E.d.M.G., B.A. and M.B.; supervision, R.M.F.B., E.d.M.G. and B.A.; project administration, R.M.F.B. and B.A.; funding acquisition, R.M.F.B. and B.A. All authors have read and agreed to the published version of the manuscript.

**Funding:** This research was funded by Fundação para a Ciência e Tecnologia through the FEDER (European fund for regional development)-COMPETE-QREN-EU (ref. UID/FIS/04650/2013 and UID/FIS/04650/2019); E-Field—"Electric-Field Engineered Lattice Distortions (E-FiELD) for optoelectronic devices", ref.: PTDC/NAN-MAT/0098/2020; Gemis—graphene-enhanced electro-magnetic interference shielding", Ref.: POCI-01-0247-FEDER-045939 and "Non-linear phononics: Manipulating the hidden quantum phases and dynamical multiferroicity", Ref. 2022.03564.PTDC.

**Institutional Review Board Statement:** This study did not involve humans or animals.

**Informed Consent Statement:** Not applicable.

**Data Availability Statement:** The datasets generated during and/or analyzed during the current study are available from the corresponding author upon reasonable request.

**Acknowledgments:** R.M.F.B acknowledge national funds (OE), through the FCT—Portuguese Foundation for Science and Technology (FCT), I.P., in the scope of the framework contract foreseen in the numbers 4, 5, and 6 of article 23 of the Decree-Law 57/2016 of 29 August and changed by the Law 57/2017, of 19 July.

**Conflicts of Interest:** The authors declare no conflict of interest.

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
