# Peer review of "Electrospun Microstructured Biopolymer Fibers Containing the Self-Assembled Boc–Phe–Ile Dipeptide: Dielectric and Energy Harvesting Properties"

_sustainability, doi:10.3390/su152216040_

Round 1
Reviewer 1 Report
Comments and Suggestions for Authors
Dear authors,
The manuscript is prepared according to journal MDPI-Sustainability.
Aim of the research was to fabricate composite microfibers using the electrospinning technique, incorporating the linear chiral dipeptide Boc-Phe-Ile into a polymeric matrix comprising different base polymers, namely PLLA and PMMA. The research on hybrid biomaterials engineered through electrospinning, incorporating dipeptide Boc-L-phenylalanyl-L-Isoleucine into biocompatible microfibers, demonstrated enhanced pyroelectric and piezoelectric properties, with Boc-Phe-Ile@PLLA microfibers exhibiting the highest piezoelectric coefficient at 56 pC/N. These findings offer insights into the potential use of polymer fibers as efficient piezoelectric energy harvesters, making them promising candidates for applications in portable and wearable devices.
The research is well-prepared, employing clear and systematic methods throughout the study.
1. The abstract provides aims and important findings of the research.
2. Introduction: Introduction provides sufficient background and includes all relevant references.
3. Materials and methods: How does the process of functionalization, as described in Scheme 1, affect the chemical structure of the Boc-protected dipeptide (Boc-Phe-Ile)?
4. Conclusions: add limitations in the preparation of the biomaterials prepared in this research.
Author Response
Dear Reviewer
Thank you very much for your comments and suggestions that gave us the opportunity to improve our manuscript. Please find our answers to your questions.
“Aim of the research was to fabricate composite microfibers using the electrospinning technique, incorporating the linear chiral dipeptide Boc-Phe-Ile into a polymeric matrix comprising different base polymers, namely PLLA and PMMA. The research on hybrid biomaterials engineered through electrospinning, incorporating dipeptide Boc-L-phenylalanyl-L-Isoleucine into biocompatible microfibers, demonstrated enhanced pyroelectric and piezoelectric properties, with Boc-Phe-Ile@PLLA microfibers exhibiting the highest piezoelectric coefficient at 56 pC/N. These findings offer insights into the potential use of polymer fibers as efficient piezoelectric energy harvesters, making them promising candidates for applications in portable and wearable devices.”
The research is well-prepared, employing clear and systematic methods throughout the study.
- The abstract provides aims and important findings of the research.
- Introduction: Introduction provides sufficient background and includes all relevant references.
- Materials and methods: How does the process of functionalization, as described in Scheme 1, affect the chemical structure of the Boc-protected dipeptide (Boc-Phe-Ile)?
Author´s response: The process of functionalization, depicted in Scheme 1, results in a significant modification of the chemical structure of the Phe-Ile dipeptide. It involves the incorporation of Boc(tert-butyloxycarbonyl) group. These Boc group enhance the overall stability of the molecule and, crucially, enable further functionalization steps. Furthermore, the introduction of Boc group has a notable impact on the solubility of the dipeptide in the solvents used for preparing polymeric solutions. This increased solubility is essential for the successful preparation of the hybrid fibrous material. Additionally, the presence of Boc group significantly enhances the physical and electrical properties of the resulting material, contributing to its unique characteristics and performance.
- Conclusions: add limitations in the preparation of the biomaterials prepared in this research.
Author´s response: The only limitation in the preparation of these biomaterials relates to the choice of solvents for preparing the polymeric solution for electrospinning. This is because the selected solvent must be suitable for both the polymer and the dipeptide, as we aim for both to be dissolved in a homogeneous solution to allow self-assembled structures to form during the process.
Additionally, the following paragraph has been added to the conclusions: “It is worth noting that the choice of solvents for preparing the polymeric solution presented a limitation in the process. The selected solvent had to be compatible with both the polymer and dipeptide, ensuring that both could be dissolved uniformly. This solvent requirement was critical to allow the formation of self-assembled structures during the electrospinning process.”

Reviewer 2 Report
Comments and Suggestions for Authors
This work presents a fabrication of composite polymer (PLLA+PMMA) microfibers via electrospinning, and embed dipeptide into the polymer matrix to have novel pyroelectric and piezoelectric responses programmed into the material. The authors provide a thorough characterization of the microfibers using SEM, DLS, photoluminescence as well as dielectric spectroscopy to illustrate sizing, optical and dielectric properties of such materials system. For the pyroelectricity aspect, the authors calculated a pyroelectric coefficient with a similar magnitude of 10^-7 C/(mK^2), indicating the potential of these fibers to be integrated into thermal sensing devices. The piezoelectric response of such materials render them as energy harvester candidates to be used in portable and wearable electronics. In all, think the paper sheds light on interesting fundamental processes in these composite microfibers (e.g. increase of dielectric constant and AC electric conductivity along with temperature increase), and also quantifies figures of merit in the piezoelectricity area for its industrial application down the road. I thus recommend publishing wish minor revision - it would be helpful for the authors to expand a little bit more on Maxwell-Wagner interfacial polarization on Page 11 to further readers' understanding of the polymer physics going on.
Author Response
Dear Reviewer
Thank you very much for your comments and suggestions that gave us the opportunity to improve our manuscript. Please find our answers to your questions.
“This work presents a fabrication of composite polymer (PLLA+PMMA) microfibers via electrospinning, and embed dipeptide into the polymer matrix to have novel pyroelectric and piezoelectric responses programmed into the material. The authors provide a thorough characterization of the microfibers using SEM, DLS, photoluminescence as well as dielectric spectroscopy to illustrate sizing, optical and dielectric properties of such materials system. For the pyroelectricity aspect, the authors calculated a pyroelectric coefficient with a similar magnitude of 10^-7 C/(mK^2), indicating the potential of these fibers to be integrated into thermal sensing devices. The piezoelectric response of such materials render them as energy harvester candidates to be used in portable and wearable electronics. In all, think the paper sheds light on interesting fundamental processes in these composite microfibers (e.g. increase of dielectric constant and AC electric conductivity along with temperature increase), and also quantifies figures of merit in the piezoelectricity area for its industrial application down the road. I thus recommend publishing wish minor revision - it would be helpful for the authors to expand a little bit more on Maxwell-Wagner interfacial polarization on Page 11 to further readers' understanding of the polymer physics going on.”
Author´s response: We agree with the referee that we can detail more on the origin of the Maxwell-Wagner polarization. Maxwell-Wagner polarization arises on materials presenting an heterogeneous character that gives rise to the presence of inner dielectric boundary layers at interfacial or grain-boundaries, where charges can be blocked. When an electric field is applied, charges at interfaces move in response to it, giving rise to interfacial dipoles due to the induced charge separation. Depending on their mobility, these charges can be very far from each other so that the interfacial polarization can reach high values, as compared to other polarization mechanisms. That causes a strong increase in ε′ with decreasing frequency as observed in our samples. As such, change de corresponding sentence in the manuscript to the following:
“The real and imaginary components of the permittivity of both samples, with the different polymeric matrices, have small variation in the whole temperature interval. Nevertheless, a difference is observed in their low frequency region. For the sample Boc-Phe-Ile@PLLA, there is an evident rise in ε' as the frequency decreases. This phenomenon can be attributed to Maxwell-Wagner interfacial polarization [34,35], which arises from the presence of interfaces within the material, a consequence of the dipeptides being integrated into the fibers. In fact, Maxwell-Wagner polarization arises on materials presenting a heterogeneous character, such as in biological materials, phase separated polymers, composites, blends, crystalline or liquid crystalline polymers, multilayer films, biphasic nanostructures and others [34]. Due to their composite nature, they give rise to the presence of inner dielectric boundary layers at interfaces or grain-boundaries, from where charges can move or be blocked. As such, when an electric field is applied, charges at interfaces move in response to it, giving rise to interfacial dipoles due to the induced charge separation. These charges can become very far from each other, so that the interfacial polarization can reach high values, as compared to other polarization mechanisms [34 ]. This then causes an increase in ε' with decreasing frequency, as observed in our samples, thus confirming the successful inclusion of the dipeptides inside the nanofibers polymeric matrix.”

Reviewer 3 Report
Comments and Suggestions for Authors
1- It is recommended to analyze the physical properties (such as tensile strength) as well.
Author Response
Dear Reviewer
Thank you very much for your comments and suggestions that gave us the opportunity to improve our manuscript. Please find our answer to your question.
- It is recommended to analyze the physical properties (such as tensile strength) as well.
Author´s response: Thank you for the suggestion. However, we believe that measuring mechanical properties would not add significant information to the already extensively characterized physical properties. Additionally, we do not have the necessary equipment to perform tensile strength tests.

Reviewer 4 Report
Comments and Suggestions for Authors
The authors provide an extensive manuscript on the preparation and characterization of polymer-based nanofibers containing, in the authors' words, Boc-Phe-Ile dipeptides' microdomains. Even when the amount of work is huge, the work presents some flaws that are difficult to understand, and some results are not presented in a clear manner. In addition, some statements must be scientifically proven. I believe this work needs a major revision before being considered suitable for publication. Below you will find some points to be considered:
1) line 122: Please, provide the yield of the reaction along with the proper characterization of the molecule (FTIR, NMR). Also, it would be beneficial for the manuscript to clearly state the importance of the incorporation of the Boc groups.
2) During the preparation of blended fibers; How was the solubility of the dipeptide in DCM? And What is the final purpose of studying the self-assembly property of this dipeptide in water/aqueous media if no water is employed during the preparation of dipeptide/PLLA fibers, which also, are the main materials under study here?
3) Any explanation for the curious morphology presented in Figure 2 a3?
4) line 321, and a major concern: I'm not sure how emission spectra can confirm the preexistence of dipeptide microspheres within fibers. It could serve as a demonstration for the presence of the dipeptide, but it is not clear, at least for me, how the authors can ensure this morphologic information. This is a very important issue, since in the Title of the article the authors state the preparation of Microstructured Biopolymer Fibers while no strong evidence of a microstructured morphology is presented.
5) Also regarding the title, Biopolymer fibers is not so correct if one of the systems studied is based on PMMA. Consider this please.
6) In dielectric characterization (Figure 7), I do not understand which relaxation is observed by the authors in Boc-Phe-Ile@PMMA. I cannot see any apparent relaxation ascribed to dipolar entities (as they mentioned in the manuscript), so I'm unsure how the authors calculate an activation energy for this system. Indeed, it is rare that, in this case, no beta relaxation is observed, which is widely known for PMMA.
7) line 385, replace figure 8 by Figure 8.
8) line 396: Even when this could be true, the presence of ionic impurities should also be mentioned. Also, did the author properly dry the materials before BDS measurements? In the experimental section, they did not refer to this issue, considering that DMF is a high boiling point solvent and could remain occluded within fibers.
8) Pyroelectricity characterization: Blank experiments of bare PLLA and PMMA fiber should be incorporated in order to verify if dipeptide inclusion has a positive or negative effect on this pyroelectric behavior.
9) line 491-492: "Consequently, the incorporation of the dipeptide clearly enhances the piezoelectric response of the fibers containing the PLLA polymer." Can the authors explain what would be the effect or mechanism through which dipeptide domains within fibers help to increase the piezoelectric property?
Author Response
Dear Reviewer
Thank you very much for your comments and suggestions that gave us the opportunity to improve our manuscript. Please find our answers to your questions.
“The authors provide an extensive manuscript on the preparation and characterization of polymer-based nanofibers containing, in the authors' words, Boc-Phe-Ile dipeptides' microdomains. Even when the amount of work is huge, the work presents some flaws that are difficult to understand, and some results are not presented in a clear manner. In addition, some statements must be scientifically proven. I believe this work needs a major revision before being considered suitable for publication. Below you will find some points to be considered:”
1) line 122: Please, provide the yield of the reaction along with the proper characterization of the molecule (FTIR, NMR). Also, it would be beneficial for the manuscript to clearly state the importance of the incorporation of the Boc groups.
Author´s response: We have incorporated the requested information regarding the yield of the reaction and the characterization of the molecule, and we have highlighted the significance of the Boc group incorporation in our revised manuscript. “This functionalized dipeptide was subsequently subjected to a 24 h drying process in an oven operating at a temperature of 50 ℃, yielding 92 %, indicating high efficiency of the functionalization.
To validate the chemical structure of the Boc-protected dipeptide, Boc-Phe-Ile, nu-clear magnetic resonance (NMR) spectroscopy was performed using a Bruker Avance III 400 instrument operating at a frequency of 400 MHz for 1H. FTIR-ATR spectrum of Boc-L-phenylalanyl-L-Isoleucine powder, acquired using a Spectrum Two™ spectrophotometer from PerkinElmer, displayed a characteristic peak at 1711 cm⁻¹, confirming the presence of a carbonyl group belonging to the Boc group (see Figure S1). The 1H NMR spectrum exhibited distinct peaks at δ 1.4 ppm (singlet, 9H, C(CH₃)₃) and 5.2 ppm (broad singlet, 1H, NH Boc), confirming the successful incorporation of the Boc group (see Figure S2). The incorporation of the Boc groups is crucial, as it not only increases the stability of the molecule but also facilitates subsequent functionalization steps, as highlighted in the manuscript. Notably, it promotes the solubility of the dipeptide in the solvents used for preparing polymeric solutions and, in a significant quantity, enhances the physical and electrical properties of the hybrid fibrous material.”
2) During the preparation of blended fibers; How was the solubility of the dipeptide in DCM? And What is the final purpose of studying the self-assembly property of this dipeptide in water/aqueous media if no water is employed during the preparation of dipeptide/PLLA fibers, which also, are the main materials under study here?
Author´s response: During the preparation of blended fibers, the solubility of the dipeptide in DCM was high, especially after functionalization with the Boc group, which made the molecule more hydrophobic and, therefore, more soluble in organic solvents, including DCM. As for the purpose of studying the self-assembly property of this dipeptide in an aqueous medium, even though water was not used during the preparation of dipeptide/PLLA fibers, there are two important reasons. First, we wanted to investigate the hydrodynamic properties of the nanostructures formed by self-assembly, which is best performed through dynamic light scattering (DLS) in aqueous solutions. Second, we wished to determine the Zeta potential of the nanostructures, an important parameter that also needs to be determined in an aqueous medium. This is relevant for a comprehensive understanding of the properties and potential applications of these nanostructures in different environments, including aqueous and biological systems.
3) Any explanation for the curious morphology presented in Figure 2 a3?
Author´s response: In fact, dipeptides have been reported to self-assemble into different morphologies. What is known is that conditions like solubility, the choice of solvent, pH, etc., and the dipeptide itself are the factors that determine the different morphologies. Figure 2a3 represents the formation of hollow microspheres based on the self-assembly of the Boc-Phe-Ile dipeptide. As an example, here we refer to an article that has reported the self-assembly of diphenylalanine (FF) into unilocular hollow spheres (https://doi.org/10.1039/C5CC01554E).
4) line 321, and a major concern: I'm not sure how emission spectra can confirm the preexistence of dipeptide microspheres within fibers. It could serve as a demonstration for the presence of the dipeptide, but it is not clear, at least for me, how the authors can ensure this morphologic information. This is a very important issue, since in the Title of the article the authors state the preparation of Microstructured Biopolymer Fibers while no strong evidence of a microstructured morphology is presented.
Author´s response: We appreciate your feedback and would like to clarify that the title of the article refers to the diameters of the fibers, which are in the order of micrometers. Regarding the presence of microspheres inside the fibers, it is indeed true that we attempted to observe them using scanning electron microscopy (SEM), but unfortunately, we were unable to do so. However, I would like to emphasize that in a previous article of ours, we were able to successfully observe by SEM the nanospheres inside the fibers (doi: https://doi.org/10.1039/D1MA01022K). However, it became evident through photoluminescence (PL) emission spectroscopy that the dipeptide is present inside the fibers. Based on this information, we made the suggested change in the paragraph, removing "Boc-Phe-Ile microspheres" and replacing it with "Boc-Phe-Ile dipeptide."
5) Also regarding the title, Biopolymer fibers is not so correct if one of the systems studied is based on PMMA. Consider this please.
Author´s response: I understand your concern regarding the title. In fact, PMMA (polymethyl methacrylate) is not traditionally classified as a biopolymer since it is not derived from natural sources. However, there are contexts in which PMMA is considered a biopolymer due to its use in biomedical applications and its biocompatibility (source: https://doi.org/10.3390/polym12123061). The classification of PMMA as a biopolymer can vary depending on the context and specific application. In our article, PMMA is being used in systems related to clean and sustainable energy production, and therefore, it makes sense to include it under the label of "biopolymer."
6) In dielectric characterization (Figure 7), I do not understand which relaxation is observed by the authors in Boc-Phe-Ile@PMMA. I cannot see any apparent relaxation ascribed to dipolar entities (as they mentioned in the manuscript), so I'm unsure how the authors calculate an activation energy for this system. Indeed, it is rare that, in this case, no beta relaxation is observed, which is widely known for PMMA.
Author´s response: The relaxation that was observed for Boc-Phe-Ile@PMMA is the one that has a bend (onset of decrease) in the real part of the permittivity at ~500 kHz (figure 7c) and that appears in the imaginary component of the permittivity as an increase of ’’ with increasing frequency, having a maximum just outside the measurement window. We agree with the referee that this may not be obvious. As such, we have changed the text after line 384 to the following sentence, for clarity:
“For Boc-Phe-Ile@PLLA, in Figure 7, we can discern the presence of a peak associated with dielectric relaxation and the onset of second relaxation occurring at higher frequencies, more visible in the imaginary component of the electric permittivity. For Boc-Phe-Ile@PMMA the relaxation is marked by a peak in the real part of the electric permittivity and an increase of e’’ with increasing frequency, which indicates a maximum just outside the measurement window.”
Regarding the beta relaxation, the difficulty in its observation in these samples is due to the nanoscopic diameters of the fibres and due to their composite nature. In PMMA the beta relaxation dependence with frequency becomes somewhat broad, less intense and its peak shifts to higher frequencies when PMMA is reduced in size [Dynamics of a and b processes in thin polymer films: PVA and PMMA, PHYSICAL REVIEW E, VOLUME 64, 051807]. This is due to chain confinement that induces changes in the equilibrium conformation statistics and orientation distribution of the polymer chains [M. Wubbenhorst, C. A. Murray, J. A. Forrest and J. R. Dutcher, "Dielectric relaxations in ultra-thin films of PMMA: assessing the length scale of cooperativity in the dynamic glass transition," Proceedings. 11th International Symposium on Electrets, Melbourne, VIC, Australia, 2002, pp. 401-406, doi: 10.1109/ISE.2002.1043027.]. Since the chain confinement results in more elongated chains in the nanofibers, the local molecular packing is improved which tends to speed-up the beta-process [Dynamics of a and b processes in thin polymer films: PVA and PMMA, PHYSICAL REVIEW E, VOLUME 64, 051807; M. Wubbenhorst, C. A. Murray, J. A. Forrest and J. R. Dutcher, "Dielectric relaxations in ultra-thin films of PMMA: assessing the length scale of cooperativity in the dynamic glass transition," Proceedings. 11th International Symposium on Electrets, Melbourne, VIC, Australia, 2002, pp. 401-406, doi: 10.1109/ISE.2002.1043027]. From the literature, activation energy values for bulk PMMA give Ea ~ 0.81 eV [Relaxation Studies in PEO/PMMA Blends, Macromolecules 2000, 33, 1002-1011], but its value can reach as low as 0.34 eV in thin PMMA films [Dynamics of a and b processes in thin polymer films: PVA and PMMA, PHYSICAL REVIEW E, VOLUME 64, 051807]. These values are similar to the Ea = 0.53 eV that we obtained in our Boc-Phe-Ile@PMMA nanofibers. As such, we agree with the referee that we cannot rule out the presence of the beta relaxation in our results. Thus, in pages 13-14 we placed the following text:
“On the other hand, the glass transition temperature for PMMA is ~378 K [41], which is substantially higher than in PLLA and is beyond the measured temperature region. In this respect, in bulk PMMA for T < Tg, the behaviour of its electric permittivity is influenced by the b-relaxation. This relaxation in PMMA arises from the rotation of the pendant side chain (-COOCH3) around the carbon-carbon bond linking the side chain to the polymer backbone [42]. In PMMA it is known that the beta relaxation dependence with frequency becomes somewhat broad, less intense and its peak shifts to higher frequencies when PMMA is reduced in size [43]. This is due to chain confinement that induces changes in the equilibrium conformation statistics and orientation distribution of the polymer chains [44]. Since the chain confinement results in more elongated chains in the nanofibers, the local molecular packing is improved which tends to speed-up the beta-process [43,44]. From the literature, the b-relaxation activation energy for bulk PMMA is Ea ~ 0.81 eV [45], but its value can reach as low as 0.34 eV in thin PMMA films [43]. These values are similar to the Ea = 0.53 eV that we obtained in our Boc-Phe-Ile@PMMA fiber mats indicating the presence of the b-relaxation in the fibres due to the PMMA polymeric matrix. Additionally, as the movement of the ionic species in the polymer is hindered, the obtained activation energy can also reflect the behaviour of the dipeptide inclusions. Namely, it can also be due of the ionic transitions of the carboxylic acid group at the termination of the dipeptide molecule.”
7) line 385, replace figure 8 by Figure 8.
Author´s response: Corrected. Thank you!
8) line 396: Even when this could be true, the presence of ionic impurities should also be mentioned. Also, did the author properly dry the materials before BDS measurements? In the experimental section, they did not refer to this issue, considering that DMF is a high boiling point solvent and could remain occluded within fibers.
Author´s response: Thank you for the comment. However, the fibers were completely dried after the electrospinning process. Inherent to the electrospinning process is the in-flight drying of the solution jet, which is why the fibers are formed by this technique (otherwise droplets, instead of fibers, are produced). Additionally, after preparation the samples are themselves, oven dried before measurement, ensuring the proper sample preparation for the dielectric measurements.
8) Pyroelectricity characterization: Blank experiments of bare PLLA and PMMA fiber should be incorporated in order to verify if dipeptide inclusion has a positive or negative effect on this pyroelectric behavior.
Author´s response: Both polymers are not pyroelectric, because they do not crystallize in any polar crystallographic point group. Please see the references: (https://doi.org/10.1021/ma001630o and https://doi.org/10.1016/0032-3861(94)90224-0). Therefore, they do not make any contribution to the pyroelectric coefficient measured, the response is due only to the dipeptide embedded into the polymer fibres.
9) line 491-492: "Consequently, the incorporation of the dipeptide clearly enhances the piezoelectric response of the fibers containing the PLLA polymer." Can the authors explain what would be the effect or mechanism through which dipeptide domains within fibers help to increase the piezoelectric property?
Author´s response: To understand the mechanism how the dipeptide influences the piezoelectric properties of the hybrid fibres, one had to know the dipeptide crystal structure. This is not known. We are attempting to grow sizeable crystals to be able to solve their structure. However, it is difficult to grow good dipeptides crystals suitable for X-ray diffraction.

Round 2
Reviewer 4 Report
Comments and Suggestions for Authors
The authors have solved adequately my observations. I think the manuscript now is able to be considered for publication